# The Effect of Leaf Wounding on Basil Plants of Different Developmental Stages

**DOI:** 10.3390/plants11202678

**Published:** 2022-10-11

**Authors:** Nikolaos Konstantis, Konstantinos Koskorellos, Areti Balou, Athina Paravolidaki, George Garantziotis, Christina Eleni Koulopoulou, Athanasios Koulopoulos, George Zervoudakis

**Affiliations:** 1Department of Agriculture, University of Patras, 27200 Amaliada, Greece; 2Department of Business Administration of Food & Agricultural Enterprises, University of Patras, Seferi Str. 2, 30100 Agrinio, Greece; 3Department of Chemistry, National and Kapodistrian University of Athens, 15784 Athens, Greece; 4Department of Biosystems and Agricultural Engineering, University of Patras, 30200 Mesolongi, Greece

**Keywords:** anthocyanins, chlorophyll, photosynthesis, stomatal conductance, stress, transpiration

## Abstract

Leaf wounding is a common stress that triggers a great number of plant mechanisms, while the overall plant status and age could also be critical for these mechanisms. However, there are not sufficient data about plants’ physiological responses after leaf wounding that has been imposed at different developmental stages. In this study, physiological parameters, such as photosynthesis, transpiration, and stomatal conductance, as well as the chlorophyll and anthocyanin leaf contents, of *Ocimum basilicum* var. *minimum* L. plants were measured for seven days on wounded plants during three different developmental stages (vegetative, budding, and flowering). All of the measurements were conducted on control and wounded plants, while on the latter they were conducted on both wounded and intact leaves. The physiological parameters mentioned above revealed a remarkable decrease in wounded leaves of the budding and flowering plants, while they seemed to be only partially affected on the leaves of vegetative plants. The physiological parameters’ decrease was not only an immediate plant response that was observed 1–2 h after wounding, but, in general, it was constant (during the seven days of treatments) and diurnal (from 8 a.m. to 8 p.m.). The wounded leaves revealed an immediate and constant anthocyanin content decrease during all of the developmental stages, while the corresponding chlorophyll decrease was mainly evident in the flowering plants. Regarding the intact leaves, they exhibited, in general, a similar profile to that of the control ones. The results above reveal that at the vegetative stage, basil plants are more tolerant to leaf wounding than those at the budding and flowering stages, implying that the plant’s response to wounding is a phenomenon that depends on the plant’s developmental stage.

## 1. Introduction

Leaf wounding is a common stress for plants, and it is caused either by herbivore animals (biotic) or by mechanical forces, such as strong wind, hail, or rain (abiotic) [1]. Mechanical or herbivory wounding causes injuries or tears and results in localized cell death, loss of water and solutes, and disruption of the vascular system [2]. Consequently, it affects a wide range of physiological processes, such as photosynthesis, transpiration, and respiration [3,4]. Moreover, leaf injury, even that caused by a mechanical force, not only leads to structural and physiological damage, but can also be a potential threat as a gateway for pathogen invasion [5]. Thus, plants have developed a wide range of constitutive defense mechanisms to cope with leaf wounding [6,7], and many plant species can distinguish insect herbivory damage from mechanical damage by using the chemical cues generated by the insects [8]. Localized tissue damage elicits biochemical and molecular responses, including enzyme activation, gene and metabolic regulation [3,7], and phytochemical reactions concerning several secondary metabolites [9]. All of these lead to quick healing and closure of the wounded tissues or even the activation of mechanisms such as programmed cell death [7]. Furthermore, leaf wounding induces the synthesis of signaling molecules, such as reactive oxygen species (ROS) [6,10] or jasmonates, which are known regulators of plant defense and contribute to the communication of the damage status with either leaf-to-leaf or long-distance signaling, thus inducing systemic responses against leaf wounding [11,12]. In addition, leaf-wounded plants may also synthesize volatile organic compounds that either facilitate within-plant and plant-to-plant communication, usually inducing the enhancement of the neighboring plants’ defense [5], or even attract parasitic and predatory insects that are natural enemies of the herbivores [13].

Plant aging is a progressive process from seed development until senescence, and the plant’s lifespan depends on many factors and stresses [14]. As the main photosynthetic organs, leaves are extremely important for plant growth and development. Consequently, a leaf’s biochemical and physiological transitions during a plant’s aging contribute to the visible phenotypic changes in the whole plant [15]. In addition, during aging, plants develop alterations that lead to decreased photosynthetic capacity and growth rate [16], while the reduction of the photosynthetic potential becomes more rapid after the induction of senescence [17]. Abiotic stresses, like many other stresses, induce overproduction of ROS, which initiates the degradation of ribulose-1,5-bisphosphate carboxylase/oxygenase (RuBisCO) and other chloroplastic proteins, inducing premature plant senescence and leading to loss of crop yields [18]. In addition, aging is marked by the enhanced accumulation of abnormal proteins, resulting in dysfunction of the entire organism, as well as protein carbonylation, which is also caused by the action of ROS. Evidently, this seems to play a key role in the regulation of aging [14].

*Ocimum basilicum* L. (basil) is one of the most common annual herbs belonging to the Lamiaceae family. It is a native of Africa, India, and Asia, and it is cultivated in temperate climates throughout the world. Basil is used in traditional medicines, and it is a popular herb in the North American and Mediterranean diets. It is widely used in the food and cosmetic industries, as it contains phytochemical constituents with different pharmacological activities [19,20]. *Ocimum basilicum* var. *minimum* L. (“Greek basil”) is a common local cultivar in Greece; it is also known as “fine-leaved” basil because of its small and narrow leaves, and it is mainly used as an ornamental plant in pots and gardens [21]. In this study, the photosynthetic and transpiration rate, the stomatal conductance, and the leaf chlorophyll and anthocyanin content were investigated in both control and wounded Greek basil plants. Moreover, in the wounded plants, we measured all of the previously mentioned variables in both damaged and intact leaves.

Since previous relevant research is limited, our aim was to evaluate the effects of leaf wounding on the physiological status of Greek basil at different developmental stages and to evaluate different plant responses to stress.

## 2. Results and Discussion

Despite the observed differences between the examined developmental stages of the Greek basil plants, in general, the wounded leaves exhibited a decrease in the photosynthetic rate compared to both the intact and control leaves (Figure 1). A decrease in carbon assimilation in the remaining leaf tissues after mechanical or herbivory wounding was revealed more often than an increase or a lack of change [22], although the plant species may be important for the type of photosynthetic response [4]. The wound-induced decrease in the photosynthetic rate was not consistent throughout the vegetative stage (Figure 1a), but it was unambiguous when wounding was imposed during budding (Figure 1b), and especially during the flowering stage (Figure 1c). In particular and considering all of the days on which the measurements took place, the decreases in the wounded leaves’ photosynthetic rate compared to that of the control were, on average, 11%, 22%, and 33% for the vegetative, budding, and flowering stages, respectively. Furthermore, the decrease in photosynthetic rate was caused immediately, since it was revealed only 1–2 h after the wounding treatment (day zero). In the vegetative plants, the photosynthetic decrease was eliminated on day 2 and reappeared on day 3, while it was constant during both the budding and flowering wounding treatments. The aforementioned findings in the vegetative plants are in accordance not only with previous results on vegetative wounded basil plants [4], but also with findings on *Arabidopsis thaliana* (L.) Heynh. plants that were six weeks old, which presented a significant decrease 2 h after leaf wounding, but photosynthesis increased to the control level 24 h after wounding [3]. On day zero, the decreases in the photosynthesis of the vegetative plants’ wounded leaves were only 7% and 17% compared with the rates in the intact and control leaves, respectively. On the other hand, the day-zero decreases were 37% and 26% for the budding plants and 44% and 58% for the flowering plants, respectively. After leaf wounding, the suppression of the photosynthetic activity was due to both direct effects, such as removal of leaf area and rupture of the photosynthetic tissue [22], and indirect impacts on the remaining photosynthetic leaf tissue via several mechanisms, such as severed vasculature and altered sink demand [23].

Regarding the photosynthetic profiles of the intact leaves, they are rather similar to the profiles of the control plants during both the vegetative and the budding treatments, while they ranged between the wounded and the control ones during the flowering wounding treatment. In particular, the photosynthetic rate of the intact leaves was not decreased compared to the control rate at the vegetative and budding stages, while it was decreased by an average of 20% at the flowering stage. It is known that mechanical wounding induces respiration in plant tissues, presumably providing energy for the increased demands for defense and repair mechanisms and, consequently, altering the plant’s primary carbon metabolism. Moreover, carbon skeletons are also required for the synthesis of new molecules [2,24]. Considering that sucrose and starch are the most common substrates for accomplishing these requirements, the intact leaves of wounded plants may operate as source organs, showing an increased photosynthetic rate not only compared to that of the wounded leaves, but sometimes even in comparison with the leaves of control plants [4]. Thus, in certain developmental stages, intact leaves may fulfill the energy and carbon demands of the wounded leaves and demonstrate increased sink strength, especially in the vicinity of the wounded tissue.

The diurnal measurements of the photosynthetic rate (Figure 2) were in accordance with the aforementioned results. During the first day of the vegetative-stage treatment, the photosynthetic rate of the wounded leaves was almost similar to those of both the control and the intact leaves until about 12 p.m., but afterwards, it was decreased (Figure 2a); however, on the sixth day, it was diurnally decreased (Figure 2b). Similarly, the budding-stage measurements of both the first (Figure 2c) and the sixth (Figure 2d) day presented a decreased photosynthetic rate for the wounded leaves compared not only to the control leaves, but also to the intact leaves. At the same time, the diurnal photosynthetic rate measurements of the intact leaves resembled those of the control not only during both of the developmental-stage treatments (vegetative and budding), but also during both days of the measurements (first and sixth).

Furthermore, the control plants exhibited a gradual decrease in the photosynthetic rate during plant aging, regardless of the wounding effect, implying that the photosynthetic activity was generally decreased during aging, as has been reported by other researchers [25]. In particular, during all days on which the measurements took place, the control flowering plants showed a 40–75% decrease in the photosynthetic rate compared to the corresponding vegetative ones, while the control budding plants showed intermediate photosynthetic rate values. It is known that changes in the photosynthetic potentials of non-senescent mature leaves occur at a slow rate compared to the rapid reduction of photosynthesis after the induction of leaf senescence [17], while at the whole-plant level, as plants age, they progressively accumulate alterations that lead to reduced leaf photosynthesis and growth rates [16,25].

Regarding the transpiration rate, and considering all the measurement days, the wound-induced decreases were, on average, 9%, 14%, and 32% during the vegetative-, budding-, and flowering-stage treatments, respectively (Figure 3), compared to the control plants. A similar pattern was exhibited for stomatal conductance (Figure 4), with corresponding average decreases of 23%, 19%, and 36%. Furthermore, as was also shown for the photosynthetic rate, the intact leaves of the wounded plants revealed transpiration and stomatal conductance similar to those of the control plants for both the vegetative and budding stages, while they were decreased by an average of 27% and 28%, respectively, during the flowering-stage treatment. The results of the vegetative-stage treatment are in accordance with previous findings in vegetative basil plants [4].

Regarding day zero, similarly to the results for the photosynthetic rate, both the transpiration rate and stomatal conductance of the wounded leaves developed significant decreases during all three treatments. The day-zero decreases were 25%, 14%, and 45% for transpiration and 52%, 26%, and 52% for stomatal conductance, respectively, for the three different stages. The immediate wound-induced decrease in these physiological parameters has also been described by other researchers [4,26,27]. It is quite interesting that during the vegetative-stage treatment, the day-zero decrease was eliminated until the second day and reappeared afterwards, as was also shown previously [4,27]. On the other hand, during the budding and flowering stages, the day-zero decrease was maintained throughout the experiment, although until the second day, the decrease was partially diminished compared to that in the control plants. The aforementioned acute decreases in the gas-exchange parameters imply that: (a) A stomatal closure related to water loss from the wounded tissue may have caused photosynthetic reduction, although it has also been reported that another possible cause may be the depressed light reaction activity in the mesophyll [28], and (b) except for the wounded tissues, which were detached or dead and probably caused a permanent reduction of photosynthesis, the adjacent healthy leaf regions may have induced an extra, but transient, loss of photosynthetic rate [2].

Upon completion of the experiment, the effect of wounding on the height and the total fresh and dry masses of shoots plus leaves of the wounded plants of all of the developmental stages were unaffected compared with the corresponding control values (data not shown), suggesting that the applied wounding intensity was not quite severe enough to affect the productivity of Greek basil.

The physiological parameters revealed quite similar profiles during the treatment at each developmental stage (Figure 1, Figure 3 and Figure 4), exhibiting higher values on all kinds of leaves (wounded, intact, control) during the vegetative-stage treatment compared with the budding- and flowering-stage treatments. Moreover, the values of the physiological parameters were even higher until the third or fourth day of the vegetative-stage treatment, and afterwards, they decreased, but this profile did not appear during the budding or the flowering stage. In general, these findings are in accordance with results from other plants whose leaves were wounded during the vegetative stage, which either exhibited an initial period of a few days of high values for all kinds of leaves, after which they were decreased, or their decrease was constant from the beginning of the wounding treatment [4], thus implying that the early or late phase of the vegetative stage may be critical for physiological processes such as photosynthesis, transpiration, etc. In addition, the diurnal measurements of the transpiration and stomatal conductance were in accordance with the corresponding photosynthetic measurements (Figure 2) during both the vegetative and the budding stage (data not shown).

All of the aforementioned results show that the leaf-wounding treatment, independently of Greek basil’s developmental stage, induced a similar pattern in all of the studied physiological parameters. Regardless of the treatment or the plant developmental stage, the strong positive correlation (r ≥ 0.54, Figure 7) among photosynthesis, transpiration, and stomatal conductance according to the Pearson coefficient analysis supports this point of view.

The chlorophyll content of the wounded leaves of Greek basil seemed to be rather unaffected during the vegetative-stage treatment (Figure 5a). These results are in accordance with previous findings, when during a long-period (of about 40 days) wounding treatment for vegetative basil plants, the chlorophyll content of the wounded leaves presented a similar profile to that of the control until the ninth day after wounding, while from the 12th day, the plants developed a decrease in chlorophyll [4]. Similar results were obtained during the budding-stage treatment (Figure 5b), while during the flowering stage, a wound-induced decrease in the chlorophyll content even appeared from day zero and remained throughout the whole experiment (Figure 5c). Considering all of the measurement days, the decrease was, on average, 22%. A similar pattern was indicated for the intact leaves of the wounded plants, since they were unaffected during both the vegetative and the budding stage, while they developed a decrease of, on average, 13% during the flowering stage. It seems that the leaf chlorophyll content (Figure 5) was not correlated with the photosynthetic rate (Figure 1), as their r values (−0.14 ≥ r ≥ 0.46, Figure 7) also implied.

Several studies demonstrated that leaf chlorophyll content is decreased after wounding, but this effect depends on the wounding intensity [3,29,30]. On the other hand, it has been reported that the contents of all photosynthetic pigments (chlorophylls a and b and carotenoids) may remain unchanged or even increase depending on the type of wounding (hole punching or piercing) [31]. Apart from the type and the extent of wounding and the inherent properties of the plant species, the effect of wounding on leaf chlorophyll content seems to be related to the environmental conditions, since it has been reported that under dark conditions, wounding may delay the loss of chlorophyll [32].

Regardless of the wounding effect, the control plants exhibited a gradual decrease in chlorophyll content during aging (Figure 5). Considering all of the measurement days, the flowering control plants showed a 25–36% decrease in chlorophyll compared to the corresponding vegetative plants, while the budding control plants showed intermediate chlorophyll values. It is known that in herbaceous annual plants, the age-related photosynthesis decrease begins at the early phase of leaf senescence and proceeds with the degradation of chlorophylls [33]. This degradation is massively carried out during the last stage of plant aging and results in leaf yellowing, one of the most obvious visible signs of plant senescence [34].

The anthocyanin content of the wounded leaves exhibited a remarkable decrease compared to that of both the control and intact leaves (Figure 6). This decrease appeared already on day zero and remained constant during all three age treatments. The intact leaves presented a similar content to that of the control during the vegetative (Figure 6a) and budding stages (Figure 6b), while during the flowering stage, they presented intermediate values with respect to the wounded and the control leaves (Figure 6c). Except for the acute and immediate wound-induced decrease in anthocyanins, the corresponding gradual decrease therein in all types of leaves (wounded, intact, control) that was observed either between the developmental stages or during the seven-day treatments seemed to be the result of the plant’s aging.

It is known that anthocyanins are water-soluble phenolic compounds that are primarily responsible for the red and blue pigmentation in plants. Anthocyanins are associated with several plant functions, ranging from contributions to plant resistance against different biotic or abiotic stresses to facilitating growth, development, and reproduction [35,36]. Although anthocyanins show strong antioxidant properties and scavenge a large range of reactive oxygen species, some researchers assume that they may be of secondary antioxidant importance considering that anthocyanins accumulate in the vacuole and not in the mitochondria or the chloroplasts, where the reactive oxygen species are mainly generated [9]. However, anthocyanins are not stable after entering vacuoles because some of them degrade to some extent due to abiotic stresses. For example, hydrogen peroxide produced in mitochondria under different environmental stresses can enter the vacuole through the tonoplast. Peroxidases located in vacuoles can use anthocyanins as a substrate, oxidizing them and inducing the loss of their color, thus assisting plants in resistance to external injuries [37]. It is known that H_2_O_2_ has been found to increase within 1 h and reach a peak 3–6 h after wounding in either wounded leaves or the unwounded leaves adjacent to wounded ones [38,39]. Consequently, under wounding stress, the day-zero decrease in the leaf anthocyanin content (Figure 6) could be a result of the increased respiration and the overproduction of ROS (e.g., hydrogen peroxide), which is finally scavenged in the vacuole with the simultaneous oxidation of anthocyanins.

Regardless of the immediate response on day zero, it is known that leaf wounding often causes biosynthesis and accumulation of anthocyanin [9,36,40], which probably contributes to phytochemical responses against insect herbivory or pathogen infection. Nevertheless, the plant anthocyanin response against leaf wounding may differ between different species, leading to reduced anthocyanin levels [4], since the quantities and the types of the anthocyanins produced are greatly affected by the inherent genetic factors of the particular plant species or variety and the environmental conditions [41]. It seems that the plant anthocyanin response even differs between different cultivars, as is revealed by comparing the results for the vegetative stage from *Ocimum basilicum* var. *minimum* L. (Figure 6) with the previous results from *Ocimum basilicum* L. [4]. These findings are consistent with previous ones, which demonstrated that the cultivar has a significant influence on the total anthocyanin levels, since a specific basil cultivar may even have an anthocyanin content that is double that of another cultivar [35].

The developmental stage is another factor affecting the anthocyanin content, considering that its biosynthesis is not developed in old injured leaves [9,41]. It has been shown that the cultivar and the maturity of basil play critical roles in the herb’s anthocyanin content. The leaf anthocyanin content of different basil cultivars seemed to begin to be reduced 49 days after plant germination, although during their prior development, the cultivars exhibited different anthocyanin content profiles [41]. Our results are consistent with these findings, since a gradual decrease in the anthocyanin content—not only for the wounded and intact leaves, but also for the control leaves—was presented for Greek basil during its aging (Figure 6), indicating that the plant developmental stage and cultivar have a unique combined effect on the total anthocyanin content [41].

Regarding a possible correlation among all variables, the corresponding Pearson analysis (Figure 7) demonstrated: (a) a strong positive correlation between photosynthesis, transpiration, and stomatal conductance for all of the developmental stages and treatments and (b) a positive correlation between chlorophyll and anthocyanin content for every treatment and age, but no correlation with the intact treatment of the vegetative stage. These results are in accordance with previous findings on vegetative basil plants [4]. On the other hand, it seems that there is a weak or no correlation between the pigment content and the leaf gas-exchange parameters.

The aging and lifespan of *Ocimum basilicum* var. *minimum* L. (“Greek basil”) depend on many environmental factors and stresses, such as mechanical wounding. Our results, in agreement with several previous studies, imply that during Greek basil’s aging, the same changes as in other plants occur in plant physiology and metabolism. First of all, these changes seem to be caused in the chloroplast, leading not only to a gradual loss of chlorophyll, but also to a simultaneous and large decrease in photosynthesis, transpiration, and stomatal conductance. Moreover, the chloroplast is considered to be one of the target organelles of age-induced oxidative stress in plants. Hence, the gradual decrease in the leaf anthocyanin content during the aging of Greek basil could be the result of the increased scavenging demands for reactive oxygen species, which are mainly produced in chloroplasts and mitochondria and then translocated to vacuoles. All of the aforementioned physiological responses seem to be further affected by leaf wounding, although the developmental stage of the plant seems to be crucial for the extent of every particular response. Considering all of the measured parameters and all of the days on which the measurements took place, it seems that the younger (vegetative stage) plants were more tolerant to leaf wounding, since minor reductions were induced. These reductions were statistically insignificant (Table 1), except for the leaf anthocyanin content, which immediately and substantially decreased, thus suggesting an instant plant response against the possible wound-induced overproduction of reactive oxygen species. On the other hand, as the Greek basil plants aged, the responses were significant for all of the measured variables, especially for the older (flowering stage) plants (Table 1).

In conclusion, the findings above suggest that younger plants present a greater tolerance against leaf wounding than older ones. The existence of more effective defense mechanisms in younger plant is probably necessary in order to cope with wounding stress and, eventually, manage to complete the biological cycle. Although wounding caused a decrease in the treated leaves’ measured parameters even in the vegetative stage, it seems that the effect became systematic only during the flowering stage, in which the intact leaves exhibited decreased values, too. For a thorough understanding of the aging/wounding interaction in plants, further experimental studies are needed, since the plants’ physiological and biochemical responses are also affected by the inherent characteristics of the particular plant cultivar and the extent and type of wounding treatment.

## 3. Materials and Methods

### 3.1. Experimental Design

Forty-two young *Ocimum basilicum* var. *minimum* L. seedlings of the same age were obtained from a local nursery at the beginning of June. They were transferred to an experimental field of the University of Patras in Amaliada (southwestern Greece, 37°48′ N, 21°21′ E) and transplanted into 4 L pots filled with loamy sand soil. All plants were grown under the same environmental conditions during the whole experiment. The plants were irrigated daily in the afternoon until there was runoff from the pot. The average daily irrigation doses per plant during the 7-day treatment were 870, 1075, and 1250 mL for the vegetative, budding, and flowering stages, respectively. Fertilization was performed 5, 8, and 8 times for all plants of the three developmental stages, respectively. The fertilization dose each time was 1 gr of crystalline fertilizer 20–20–20 (N–P–K) + 5 MgSO_4_ per plant. The average daily temperature ranges during the experiment were 25.5–30.4, 25.8–28.1, and 27.5–35.8 °C, and the average daily relative humidity ranges were 54.4–67.1%, 47.9–70.5%, and 26.9–66.5% for the vegetative, budding, and flowering stages, respectively. There was no precipitation during the experiment (data recorded by a meteorological station).

After 30 days of acclimation and a growth period, the plants were divided at random into three groups of 14 plants per group, that is, one group for each developmental stage.

The wounding treatments in the three different developmental stages were conducted after 30, 50, and 65 days of acclimation and growth, that is, during the vegetative, budding, and flowering stages, respectively.

From each group, 7 plants remained intact and were used as the control for the treatment. The other 7 plants of the group were wounded (day 0) from the top to the bottom leaves of the shoot with a cork borer with a diameter of 0.5 cm, causing 1–2 holes on each leaf. Since the plants were approximately spherical, a vertical hemispherical part of each of them was chosen, and the damaging treatment was conducted only on the leaves of this half—henceforth called wounded leaves—while the leaves of the other half were the intact leaves (Figure 8).

Upon completion of each wounding treatment, 7 days after wounding, the height of the plants was measured. All of the plants were harvested, and the above-ground part (shoots plus leaves) of each one was weighed. To obtain the dry weight, the samples were dried to a constant weight in an oven at 70 °C for 72 h.

All of the experimental measurements of the leaf gas-exchange parameters and the pigment content were conducted 8 times, from day 0 to day 7, for each wounding treatment.

### 3.2. Leaf Gas Exchange

Nondestructive measurements of photosynthesis, transpiration, and stomatal conductance were obtained with a TARGAS-1 Portable Photosynthesis System (PP Systems, Amesbury, MA, USA) using an infrared gas analyzer. The measurements were obtained under natural light, temperature, and environmental CO_2_ conditions at 10:00 a.m. on wounded, intact, or control leaves that were young and completely expanded.

All of the measurements were conducted on sunny days and were obtained from all 14 plants per developmental stage. For each experimental day, two measurements were conducted on each wounded plant—one measurement of a wounded leaf and one of a corresponding intact leaf—while only one measurement was conducted on each control plant. Due to the spherical shape of the plants, peripheral leaves of the 3rd or 4th node from the top were selected, that is, leaves that had developed under full sunlight.

The leaf gas-exchange measurements on day zero (day of wounding) were conducted about 1–2 h after wounding.

Diurnal measurements of the leaf gas-exchange parameters were also conducted on the 1st and 6th day of the vegetative and budding wounding treatments. The measurements were conducted every two hours, from 8 a.m. to 8 p.m.

### 3.3. Chlorophylls and Anthocyanins

Nondestructive measurements of chlorophylls and anthocyanins were obtained with an SPAD 502DL chlorophyll meter (KONICA MINOLTA, Tokyo, Japan) and ACM-200plus anthocyanin content meter (ADC BioScientific Ltd., Hoddesdon, UK), respectively. The same sampling protocol as that described above for the leaf gas-exchange measurements was also followed for the determination of chlorophylls and anthocyanins. Leaf pigment measurements were conducted for all 14 plants per developmental stage. For each experimental day, measurements of three wounded and three intact leaves of each wounded plant and three leaves of each control plant were conducted. For each daily measurement of pigments, the leaves were selected randomly; however, they were peripheral leaves of the 3rd or 4th node from the top. A healthy leaf area was always captured and measured, even for the wounded leaves.

### 3.4. Data Analysis

The results for all the measured variables were obtained from 7 plants per group (vegetative, budding, or flowering) and treatment (wounded, intact, or control) and were plotted as the mean ± standard deviation (SD).

The plotting of the data, the Pearson correlation coefficient analysis, and all other statistical analyses were carried out with GraphPad Prism v.9.0 (San Diego, CA, USA).

## Figures and Tables

**Figure 1 plants-11-02678-f001:**
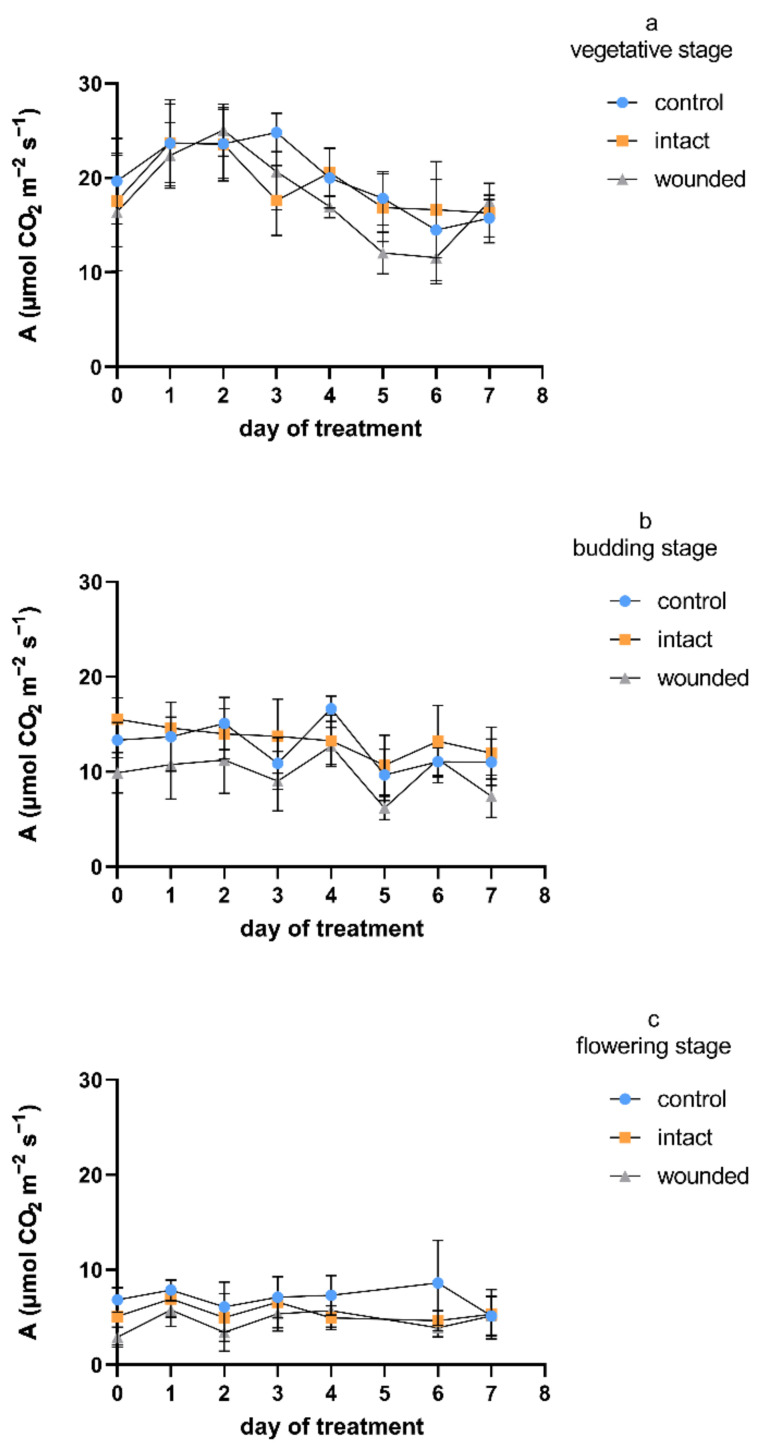
Photosynthetic rate of *Ocimum basilicum* var. *minimum* L. versus leaf wounding imposed at the (**a**) vegetative, (**b**) budding, (**c**) or flowering stage. Data are means (*n* = 7) ± standard deviation (SD).

**Figure 2 plants-11-02678-f002:**
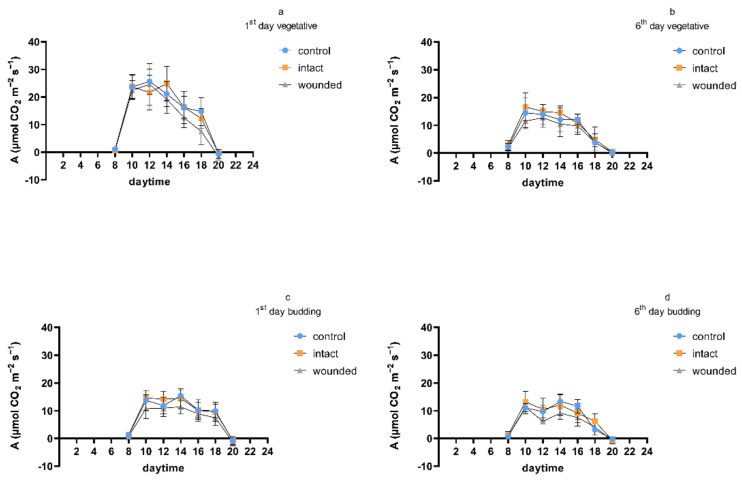
Diurnal photosynthetic rate of *Ocimum basilicum* var. *minimum* L.: one (**a**,**c**) or six (**b**,**d**) days after leaf wounding imposed at the vegetative (**a**,**b**) or budding (**c**,**d**) stage. Data are means (*n* = 7) ± standard deviation (SD).

**Figure 3 plants-11-02678-f003:**
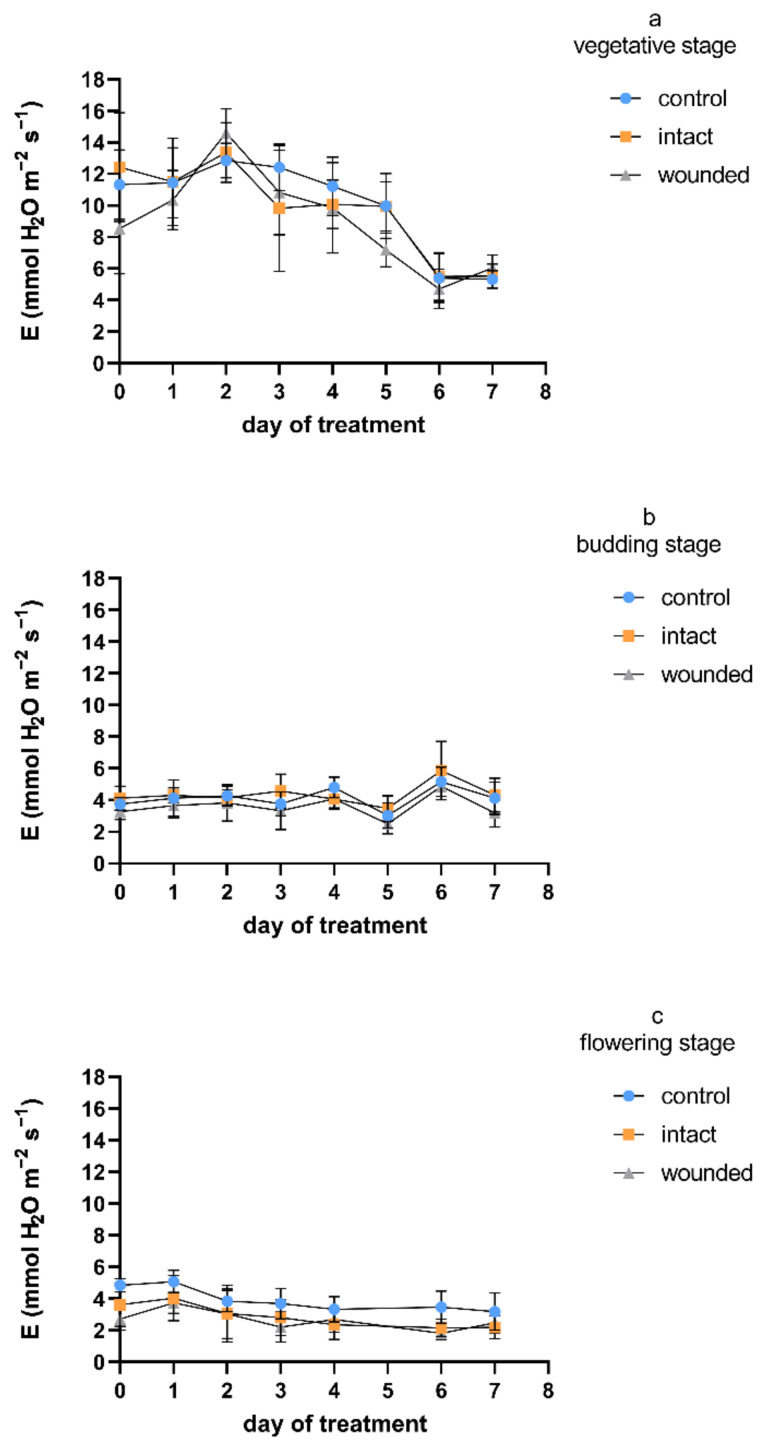
Transpiration rate of *Ocimum basilicum* var. *minimum* L. versus leaf wounding imposed at the (**a**) vegetative, (**b**) budding, or (**c**) flowering stage. Data are means (*n* = 7) ± standard deviation (SD).

**Figure 4 plants-11-02678-f004:**
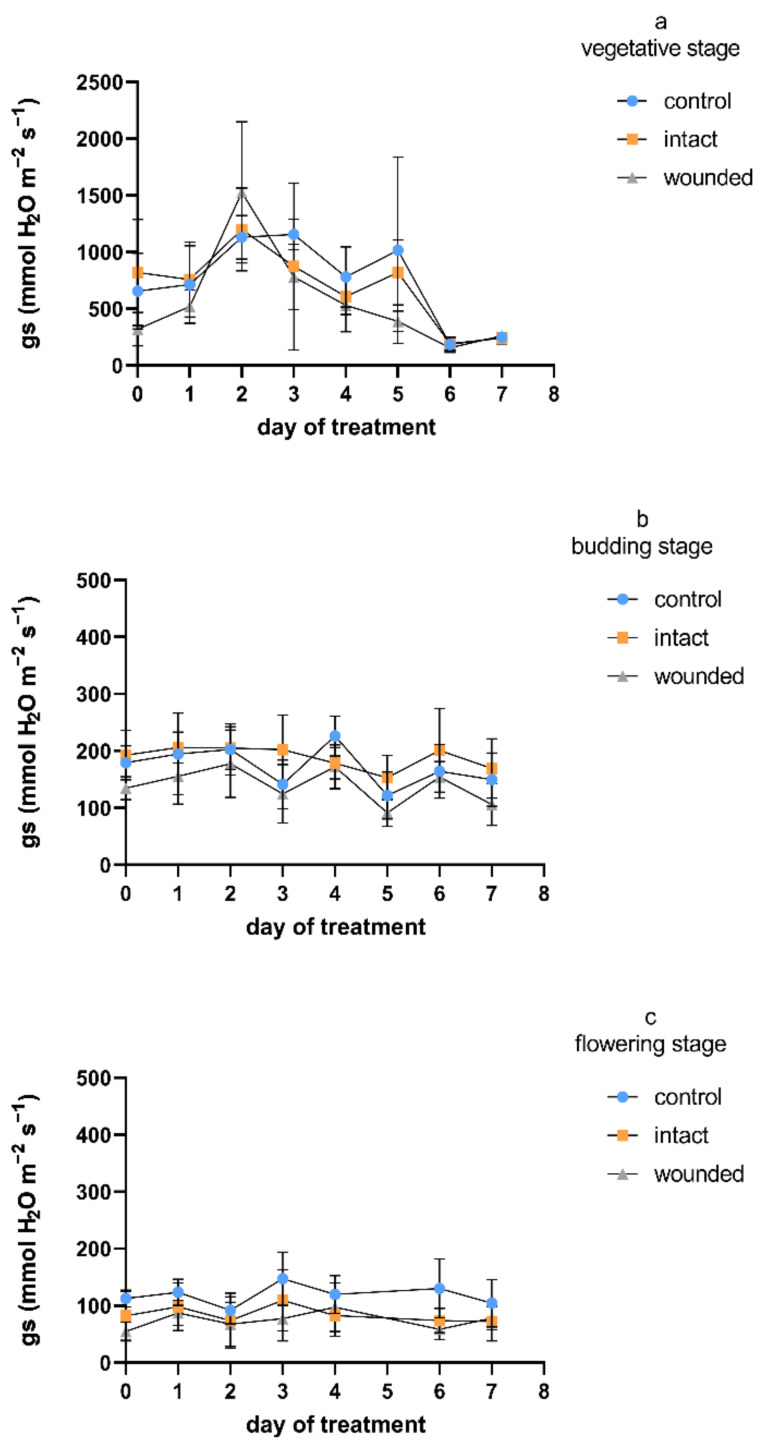
Stomatal conductance of *Ocimum basilicum* var. *minimum* L. versus leaf wounding imposed at the (**a**) vegetative, (**b**) budding, or (**c**) flowering stage. Data are means (*n* = 7) ± standard deviation (SD).

**Figure 5 plants-11-02678-f005:**
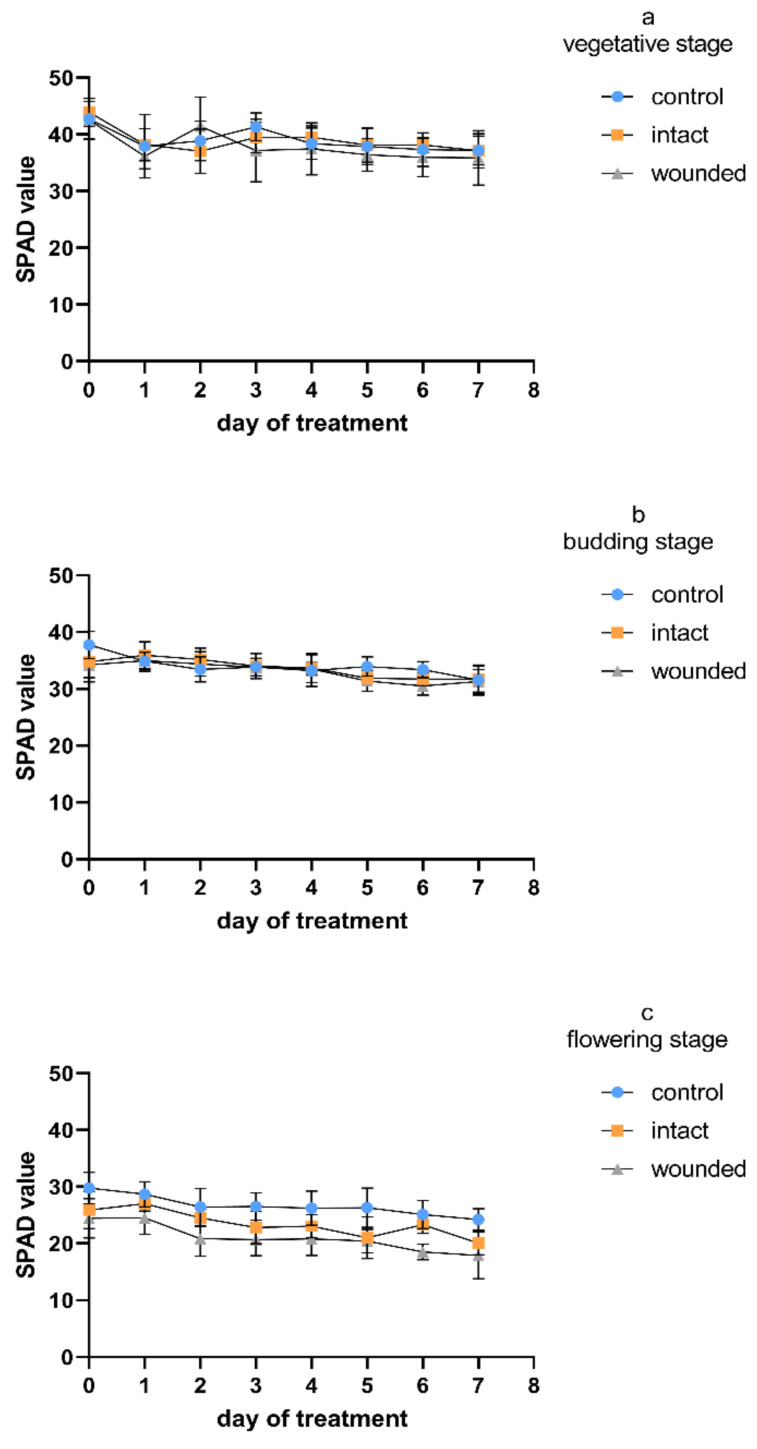
Chlorophyll content of *Ocimum basilicum* var. *minimum* L. versus leaf wounding imposed at the (**a**) vegetative, (**b**) budding, or (**c**) flowering stage. Data are means (*n* = 21) ± standard deviation (SD).

**Figure 6 plants-11-02678-f006:**
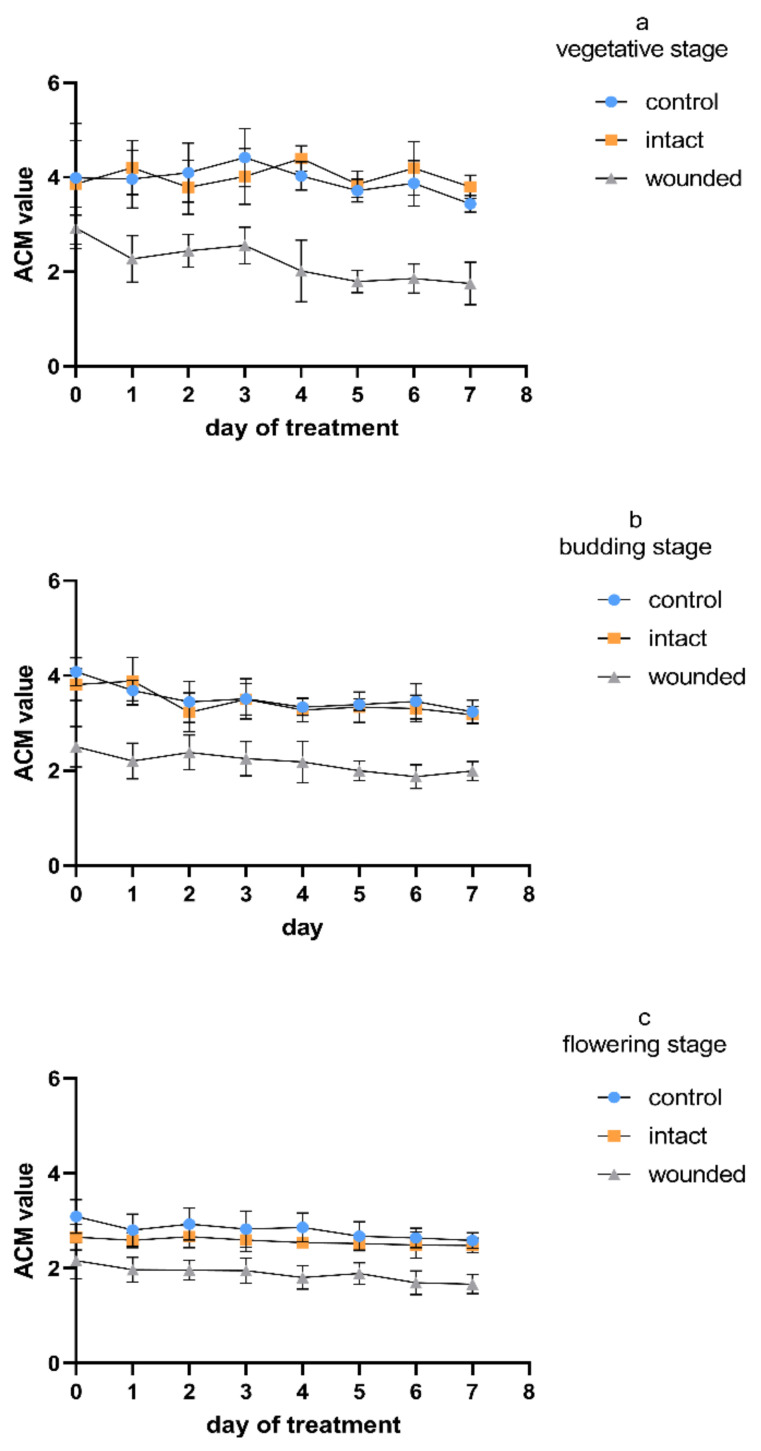
Anthocyanin content of *Ocimum basilicum* var. *minimum* L. versus leaf wounding imposed at the (**a**) vegetative, (**b**) budding, or (**c**) flowering stage. Data are means (*n* = 21) ± standard deviation (SD).

**Figure 7 plants-11-02678-f007:**
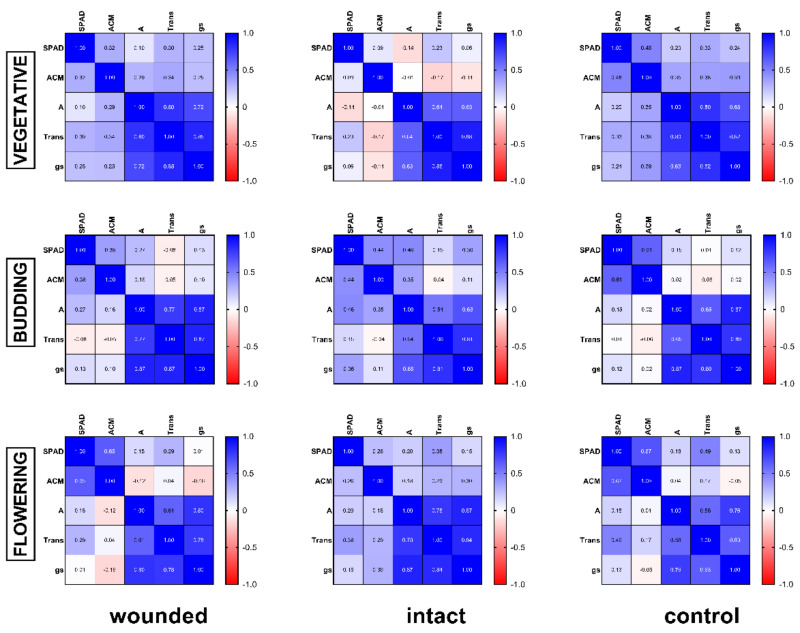
Heatmap of the Pearson coefficient analysis (r values) of the studied plant developmental stages and treatments.

**Figure 8 plants-11-02678-f008:**
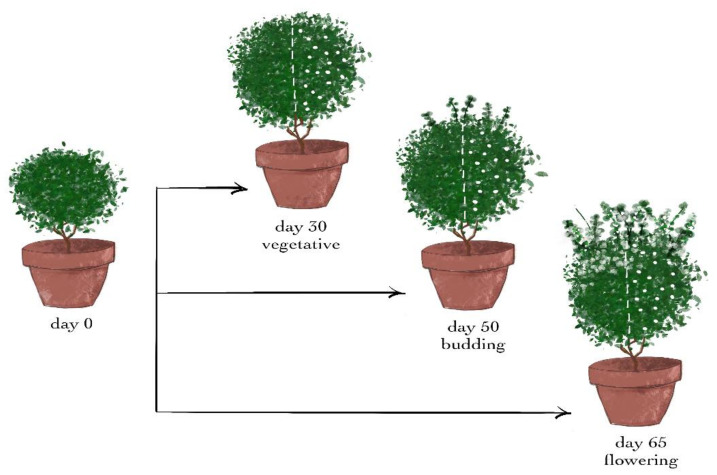
Plant treatment timeline.

**Table 1 plants-11-02678-t001:** Comparison of the treatments in different developmental stages (statistical significance according to one-way ANOVA).

Developmental Stage	Treatment	A	Trans	gs	SPAD	ACM
vegetative	control vs. intact	ns	ns	ns	ns	ns
	control vs. wounded	ns	ns	ns	ns	****
	intact vs. wounded	ns	ns	ns	ns	****
budding	control vs. intact	ns	ns	ns	ns	ns
	control vs. wounded	****	*	**	ns	****
	intact vs. wounded	****	***	****	ns	****
flowering	control vs. intact	**	****	****	****	****
	control vs. wounded	****	****	****	****	****
	intact vs. wounded	ns	ns	ns	***	****

Note: *, **, ***, and **** represent statistical significance at *p* ≤ 0.05, 0.01, 0.001, and 0.0001, respectively. ns represents statistical non-significance: *p* > 0.05.

## Data Availability

The data that support the findings of this study are available from the corresponding author upon reasonable request.

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
