# Peer review of "The Effect of Leaf Wounding on Basil Plants of Different Developmental Stages"

_plants, 2022, doi:10.3390/plants11202678_

Round 1

Reviewer 1 Report

Dear Authors,

Your manuscript needs extensive revisions and please follow recommendations and corrections in the text attached

Reviewer 2 Report

Leaf wounding caused by mechanical reason or insect herbivory, may initiate diverse biological processes, and finally affect a wide range of physiological procedures. In the manuscript, Konstantis et al characterized the leaf wounding caused physiological parameters such as photosynthesis, transpiration, and stomatal conductance, as well as the chlorophyll and anthocyanin leaf contents of different stages Ocimum basilicum var. minimum L. plants. The authors found that wounded budding and flowering leaves revealed a remarkable decrease for all the gas exchange parameters while the vegetative ones seemed to be partially affected. Interestingly, the gas exchange parameters decrease was not only an immediate plant response, in general, it was also constant and diurnal. The results above reveal that the vegetative basil plants are more unsusceptible to leaf wounding than the budding and flowering ones, implying that the plant’s response to wounding is a phenomenon dependent on the plant’s developmental stage. Overall, the manuscript presents a potentially interesting topic. Here are some minor concerns:

1.     It will be easier for the audience to follow if the authors include a plant model to show different plant ages, how to wound plants, and which leaf was selected for testing physiological parameters

2.     Most of the figures lack statistical analysis to show whether the changes are significant

3.     The stomata conductance is affected by light temperature and other environmental factors. The author should claim the specific time of a day to test physiological parameters in the Material and Methods part

Reviewer 3 Report

In the submitted manuscript, Konstantis et al., report on the physiological analysis of Basil plants following the wounding of individual leaves. Photosynthetic parameters (CO2-fixation, trabnspiration and stomatal conductance) as well as anthocyanin contents have been analysed in filed grown plants using optical devices. Based on their findings they concluded that "young" Basil plants are less suceptible to "wounding stress" compared to those of the budding and flowering stage. Although a discussion why that might be the case is missing, there are specific points which have to be addressed:

Points which have to be addressed:

- The Authors stated in the method section that:

“Upon completion of each wounding treatment, 7 days after wounding, the height of 373 the plants was measured. All the plants were harvested and the above-ground part 374 (shoots plus leaves) from each one was weighed. To obtain the dry weight, the samples 375 were dried to constant weight in an oven at 70oC for 72 h.”

I was wondering where this data is shown. Indeed, it would be interesting to know how the treatment affects plant growth and biomass production.

- It is surprising to see that the anthocyanin content declines within 1-2 hours after the leaves have been wounded with the cork borer. Considering the fact that anthocyanin biosynthesis is induced upon (a)biotic perturbations, even those leading to high levels of oxidative stress, and anthocyanins are stabile molecules, this finding is rather unexpected and needs further clarification (Although the explanation provided is already a good start). Since anthocyanin contents have been determined only indirectly, I was wondering how this measurement/ device works? Did the chamber of the device for the measurement cover the area were the leaves have been wounded (i.e., where a part of the leaf is missing)? Can the authors quantify anthocyanins spectrophotometrically using established protocols and confirm this rapid decrease of anthocyanins? Comparing wounded vs. control leaves the decrease of anthocyanins was seen only after 1-2 hours following the wounding. After that timepoint, leaves from all conditions showed the same (relative) content/changes in contents of anthocyanins. How is this explained? The same holds true for the data shown in Figure 1 (“A – photosynthetic rate”).

If indeed a sudden burst in ROS is causative for the rapid decline of anthocyanin after wounding (as suggested by the authors), incubation/infiltration of intact Basil leaves with hydrogen peroxide should also induce degradation of anthocyanins. Can the author confirm this?

- The authors need to explain more precisely, how the photosynthetic/gas exchange parameters have been obtained. For example, which light intensity was used? How was the CO2 analyzed? …

- Finally, I have difficulties to agree to the statement:

All the above findings 339 suggest that the younger plants present a greater tolerance against leaf wounding than 340 the older ones. “ (l339ff)

Photosynthetic activity (Fig 1), transpiration (Figure 3) and stomatal conductance (Figure 4) is more drastically affected in the vegetative state (4-6 days). Except for the initial drop after 2 h following wounding, in the wounded leaves these parameters paralleled the changes observed in the budding and flowering stage suggesting that they are not more sensitive to the wounding in these developmental stages. If so, one would expect that the wounded leaves would deviate more strongly from the control leaves in the analysed paranmeters in the budding and flowering stage.

- Statistics: all of the statements regarding differences between the different experimental groups are based on a semi-quantitative comparison and no statistical analysis has been performed. It is not clear to me, when a change is of relevance for the authors. It is self-evident that a statistical analysis is needed to allow a clear interpretation of the data.

- Why did the authors use the SEM instead of the SD? As far as I understood, the experiment was performed once and several data points were obtained from the different groups and treatments (one sample from the population). Hence, I would expect to see the SD to show the dispersion of the individual data points.

- Figures should be prepared using a professional program other than Excel. The authors perform some analysis with GraphPad Prism, which could be used for a better presentation of the data.

Round 2

Reviewer 1 Report

Dear authors,

Thank you for your corrections and additions on all the points raised in the previous reviewing process 

Author Response

Dear Reviewer

I would like to thank you for helping us to improve our manuscript.

Respectfully

G. Zervoudakis

Reviewer 3 Report

Dear authors

Tahnk you for the revision of the manuscript. Although you addresses fome of my comments/concernes I am still not satisfied.

1. Figures: The new figures were not provided in the revised manuscript and the figure legends were not corrected (SD instead of SEM).

2. Statistical analysis. I do not understand the table with the statistical analysis. What has been tested for in this table? The means throughout the whole kinetic? I would expect to see a detailed statistical analysis between the treatments in each developmental phase (per time point). Statistical differences should be indicated in the figure.

3. I'm still not convinced about the validity of the anthocyanin measurements. Again, the strong decline after 2h after wounding is not expected. Besides this strong decrease after 2h, the overall trend for the anthocyanin content is the same between all groups of treatments.

I therefore requested to provide more data on this finding. It is not a complicated experiment to incubate leaves (of any species) with hydrogen peroxide to show a fast decrease of anthocyanins or to determine anthocyanin contents after wounding using an extablished method (photometer).

4. Also, again, except for the anthocyanins (which only drop strongly 2h after wounding), the wounded leaves showed pretty much similar values for "A", "gs", "SPAD"/ Chl, compared to the "intact" ones (see the figures). This is the case for all developmental stages. Please explain your conclusion that the "older" plants suffer more from the wounding than the younger ones. On which parameters is this based?

Round 3

Reviewer 3 Report

.